# Acceptance and knowledge of evolutionary theory among third-year university students in Spain

**Juan Gefaell[1], Tamara Prieto[2], Mohamed Abdelaziz[3], Inés Álvarez[4], Josefa Antón[5], Juan Arroyo[6], Jose L. Bella[7], Miguel Botella[8], Anxela Bugallo[9], Vicente Claramonte[10], José Gijón[11], Emilio Lizarte[11], Rosa M. Maroto[8], Manuel Megías[1], Borja Milá[12], Cori Ramón[13], Marta Vila[14], Emilio Rolán-Alvarez[1] ***

**1** Facultade de Bioloxía, Universidade de Vigo, Campus de Vigo, Vigo, Spain, **2** Centro de Investigacións Biomédicas (CINBIO), Universidade de Vigo, Campus de Vigo, Vigo, Spain, **3** Departamento de Genética, Universidad de Granada, Avda Fuentenueva, Granada, Spain, **4** Real Jardín Botánico, Consejo Superior de Investigaciones Científicas (CSIC), Pza. de Murillo, Madrid, Spain, **5** Departamento de Fisiología, Genética y Microbiología, Universidad de Alicante, Apdo., Alicante, Spain, **6** Departamento de Biología Vegetal y Ecología, Universidad de Sevilla, Sevilla, Spain, **7** Departamento de Biología (Genética), Facultad de Ciencias, Universidad Autónoma de Madrid, Madrid, Spain, **8** Departamento de Medicina Legal, Toxicología y Antropología Física, Universidad de Granada, Parque Tecnológico de la Salud, Av. de la Investigación, Granada, Spain, **9** Departamento de Pedagogía y Didáctica, Universidade da Coruña, Campus de Elviña, A Coruña, Spain, **10** Unidad Docente de Lógica y Filosofía de la Ciencia, Departamento de Filosofía, Facultad de Filosofía y Ciencias de la Educación, Universitat de València, Valencia, Spain, **11** Departamento de Didáctica y Organización Escolar, Universidad de Granada, Campus de Cartuja, Granada, Spain, **12** Museo Nacional de Ciencias Naturales, Consejo Superior de Investigaciones Científicas (CSIC), Madrid, Spain, **13** Departament de Biología, Universitat de les Illes Balears, Palma de Mallorca, Carretera de Valldemossa, Illes Balears, Spain, **14** Departamento de Bioloxía, Facultade de Ciencias, Universidade da Coruña, Campus da Zapateira, A Coruña, Spain

* rolan@uvigo.es

**Data Availability Statement:** The data underlying the results presented in the study are available from Supplementary tables included in this submission. The main Data is available in https://

## Abstract

The theory of evolution is one of the greatest scientific achievements in the intellectual history of humankind, yet it is still contentious within certain social groups. Despite being as robust and evidence-based as any other notable scientific theory, some people show a strong reluctance to accept it. In this study, we used the Measure of Acceptance of the Theory of Evolution (MATE) and Knowledge of Evolution Exam (KEE) questionnaires with university students from four academic degree programs (Chemistry, English, History, and Biology) of ten universities from Spain to measure, respectively, acceptance and knowledge of evolutionary theory among third-year undergraduate students ($n_{MATE}$ = 978; $n_{KEE}$ = 981). Results show that acceptance of evolution is relatively high (87.2%), whereas knowledge of the theory is moderate (5.4 out of 10) although there are differences across degrees (Biology>Chemistry>History>English), and even among various universities (ranging from 4.71 to 5.81). Statistical analysis reveals that knowledge of evolutionary theory among Biology students is partially explained by the relative weight of evolutionary themes within the curriculum, suggesting that an increase in the number of hours dedicated to this topic could have a direct influence on students' knowledge of it. We also found that religion may have a significant—although relatively small—negative influence on evolutionary theory acceptance. The moderate knowledge of evolution in our undergraduate students, together with the potential

figshare.com/articles/dataset/Dataset_of_
Acceptance_and_knowledge_of_evolutionary_
theory_among_third-year_university_students_in_
Spain_/12623747.

**Funding:** This work was supported by Xunta de Galicia (ED431C 2016-037), FONDOS FEDER ("unha maneira de facer Europa"), and Spain's Ministerio de Economía, Industria y Competitividad, Agencia Estatal de Investigación (CGL2016-75904-C2-1-P), Dr. Emilio Rolán-Alvarez.

**Competing interests:** The authors have declared that no competing interest exist.

problem of acceptance in certain groups, suggests the need for a revision of the evolutionary concepts in the teaching curricula of our students since primary school.

## Introduction

Evolution is a core theory of biology [1,2], although its influence extends to other disciplines such as philosophy [3–5], psychology [6], and medicine [7,8], as well as, in general, most life and social sciences, including politics and policy [9–11]. Evolution represents a well-established and mature theory that can explain how organisms evolve and differentiate from the last universal common ancestor (usually known as LUCA): a life form that originated on our planet about 3700 billion years ago [12]. Therefore, both the notion of descent with modifications and the theory of evolution represent key pieces of knowledge in our societies to understand both the world in which we live and, as well, our place in nature [13]. On the other hand, misconceptions about evolution commonly persist along the whole educational system, an effect that has been attributed to the possibility that evolution theory is counterintuitive [14], even working against our natural reasoning [15]. Nevertheless, and perhaps due to this key relevance and difficulty of understanding, the idea of evolution has been debated since its first formal proposal by Darwin [16,17]. A pioneer study led by Miller et al. [18] has shown that the level of evolutionary acceptance in citizens can vary considerably across countries and social groups. Therefore, research on the causes of such variations, as well as means to improve acceptance and knowledge levels, can be considered one of the main subtopics of evolution education [19].

There is lack of consensus in the field about the major determinants of evolution acceptance. For some authors, obstacles to accepting evolutionary theory show particularities that differ from those relating to other well-established scientific theories [20]. Such aspects have been attributed to the fact that certain intransigent religious views clash with evolutionary views, together with the variation in religiosity observed among countries or even social groups within a given country [2,21,22]. However, empirical studies [23–25] typically have found three major factors determining evolution acceptance: religiosity, evolution understanding, and nature of science (NOS) understanding [26]. Therefore, a complex problem like this is most probably multifactorial while, at the same time, the different factors involved may presumably be partially correlated. For example, degree of religiosity is about equally correlated with evolution acceptance and two measures of education level in data from 34 countries [27]. The difficulty of determining factors leading to evolution acceptance has been suggested to hamper the derivation of optimal education strategies for our citizens [19]. In fact, it has been claimed that the level of evolution acceptance has remained low for the last 30 years in the USA [21].

One factor that could contribute to these disparate views is variability in the measures used to assess evolution acceptance: each can capture different evolution acceptance characteristics or even show differential levels of influence with other factors (reviewed in [19]). A variety of tests have been proposed to measure evolution acceptance, including the Measure of Acceptance of the Theory of Evolution (MATE) [28], the Evolutionary Attitudes and Literacy Survey (EALS) [29], the Inventory of Student Evolution Acceptance (I-SEA) [30], the Generalized Acceptance of EvolutioN Evaluation (GAENE) [31], the Attitudes toward Evolution (ATEVO) [32], etc. Although a consensus has not fully been achieved, the MATE seems to incorporate several of the ideal properties for this measurement [33]. The main advantage is that the

MATE has been the most frequently-used test to date [19,21,34,35]. Moreover, when this instrument has been compared with alternatives, a moderate to high correlation between them has typically been observed [19,21,36,37].

The same lack of consensus exists when measuring the level of evolution knowledge, given that several alternative assessment tools are available: the Conceptual Inventory of Natural Selection (CINS) [38], Conceptual Assessment of Natural Selection (CANS) [39], Knowledge About Evolution (KAEVO) [32], Assessing Contextual Reasoning about Natural Selection (ACORNS) [40] and Knowledge of Evolution Exam (KEE) [41], among others. Another potential problem in studies trying to compare results of acceptance and knowledge about evolution across countries or social groups is the high heterogeneity of data available, as different studies focus on primary/secondary school, undergraduate, graduate, teachers or even variant social groups. Moreover, in certain cases, the questionnaires are offered online without any control of the time employed by each respondent. Therefore, any comparative study, in order to properly compare different experimental groups, should ideally use the same design on a homogeneous data set when possible.

Spain is considered an industrialized country (13[th] position in the World GDP ranking) [42] belonging to the European Union. Information about the knowledge and acceptance of evolution in Spain is scarce, although supposedly there is not any special social group reluctant to embrace evolution (showing intermediate to high ranking regarding evolution acceptance in Miller et al. [18]). Nevertheless, from within the educational field, it has been claimed that the curricula of primary and secondary school is rather poor in introducing evolutionary theory, at least compared to the most (educationally) advanced countries [43]. This fact could explain the subjective feeling of many university teachers that ideas on evolution are not in the students' backpack when they reach the university. With this potential problem in mind, we tried to check the prevalence of evolution acceptance and the knowledge level regarding this topic in undergraduate students attending the Spanish university system. We presented a three-section online questionnaire (incorporating items on demographics, evolution-acceptance, and evolution-knowledge) to third-year university students from several disciplines and institutions throughout Spain. The main objective was to characterize the level of acceptance and knowledge at different universities and degree programs and identify any group particularly reluctant to accept those ideas. A secondary objective was to check if the level of evolutionary theory in each degree program curriculum could explain the variation in evolutionary knowledge across universities.

## Materials and methods

### Bibliographic review of undergraduate evolution acceptance

One advantage of the MATE test is that it has been frequently used to infer the level of evolution acceptance from different countries (52 studies reviewed in Barnes et al. [19]). However, these studies had a high degree of individual diversity; further, we revised them to exclude all cases that did not represent exclusively undergraduate university students. In addition, in November 2019, we searched references in the WEB of Science database by applying the keywords "questionnaire" and "evolution"; in the list obtained, we used exclusively those conducted with the MATE and on undergraduate university students.

### Participants

We scored certain demographic questions and evolution acceptance and knowledge tests administered to 1050 third-year students ($n$ = 981 after excluding incomplete data) from four degree programs (Chemistry, History, English Philology, and Biology) at ten Spanish

universities (Alicante, Autónoma de Madrid, Complutense de Madrid, Granada, Illes Balears, Salamanca, Santiago de Compostela, Sevilla, Valencia, and Vigo; Fig 1). Erasmus program students (foreign students funded by the European Union who came to study for one academic year) were identified and excluded from final analyses. We chose two science and two humanity degrees, having replicates within each group and assuming *a priori* that Biology students

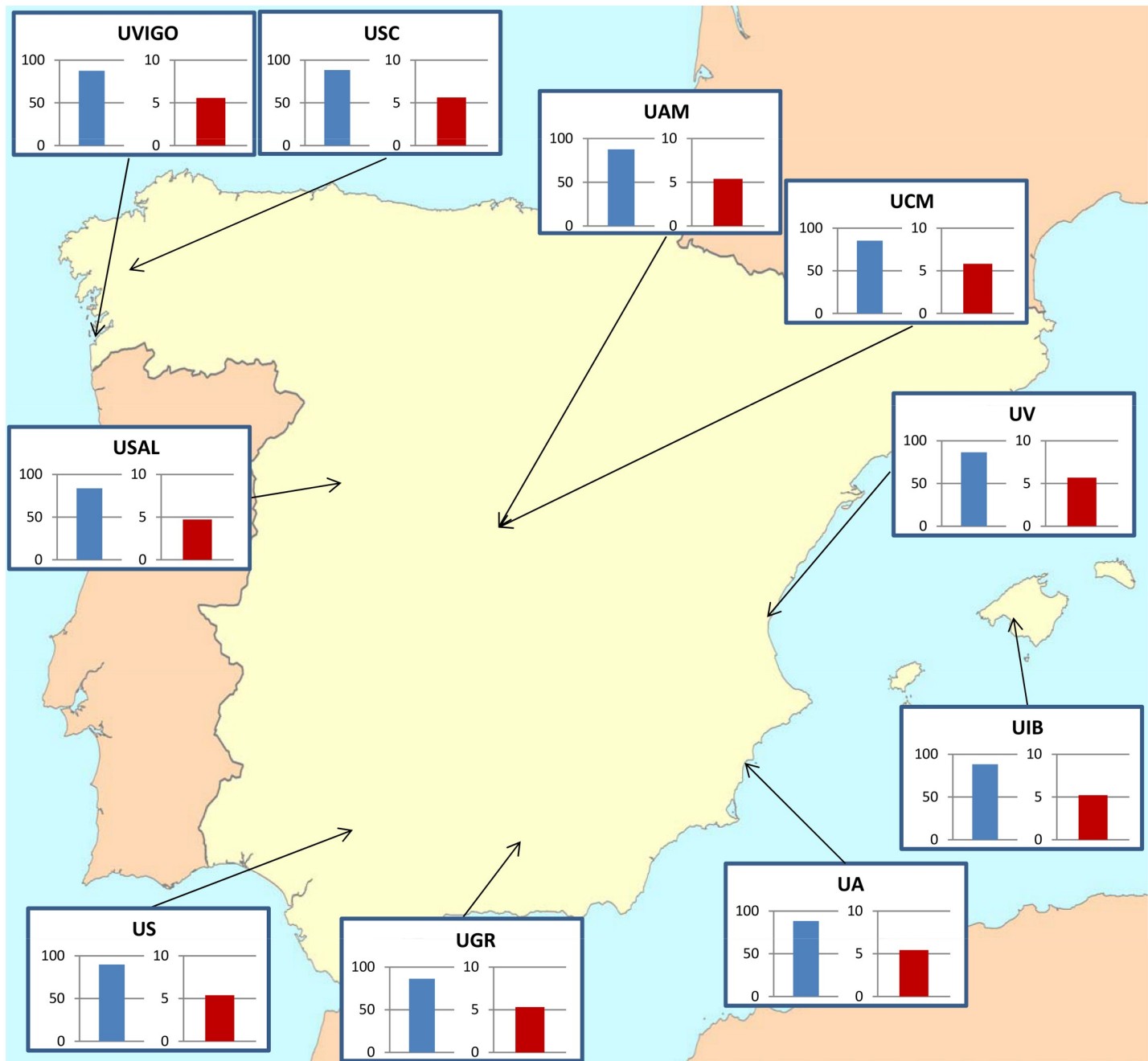

**Fig 1. Comparison of MATE and KEE across Degrees.** Bars are mean scores for MATE (in blue) and KEE (in red) for the different universities in the Spanish public university system included in the study and their geographic location. UVIGO (Universidade de Vigo), USC (Universidade de Santiago de Compostela), UAM (Universidad Autónoma de Madrid), UCM (Universidad Complutense de Madrid), UV (Universidad de Valencia), UIB (Universitat Illes Balears), UA (Universidad de Alicante), UGR (Universidad de Granada), US (Universidad de Sevilla) and USAL (Universidad de Salamanca).

should have the highest rank in at least evolutionary knowledge (as it is the only subject where evolutionary concepts appear in the curriculum). We studied 10 of 21 possible public universities (awarding these four degrees) which could be considered as a representative sample for this country. The assessment was run during the 2018–2019 academic year, between September and December; the sample size per degree and university is shown in S1 Table.

## Data collection

We asked for permission from the dean at each school and university to share our questionnaires with students. Usually, the dean suggested contacting a particular teacher of a third-year course from the school. Typically, one of the authors met with the class and explained the procedure to the students, including its voluntary and anonymous nature. Students were surveyed through a Google Forms quiz accessible through the Internet using their own electronic devices; they were given 15 minutes to finish the task (see S1 Table). The questionnaire was divided into three sections: personal-demographic questions, MATE-related questions and KEE-based questions (see next section), following the order above. A university-specific alphanumeric code was provided immediately before the start of the test; it was required in order to proceed and submit the form. We checked *a posteriori* that all questionnaires from a particular university and degree were completed on the exact dates and times, following notification from the coordinator.

The demographic questionnaire (see S1 Table) asked the sex of the participant (*Sex*; man, woman, I prefer not to choose), the age (*Age*), academic level attained (*Academic level*; secondary, graduated, doctor), secondary school itinerary (*Itinerary*; science, technology, humanities, social sciences, and art), degree that was being pursued (*Degree*; Chemistry, History, English, and Biology), university (*University*; the 10 schools described excluding Erasmus) and the religiosity level (*Religiosity*; usually attends religious services or not). The demographic questions are listed and summarized in various columns of the S1 Table.

## Tests for evolution acceptance and knowledge

We used the MATE test [28] to assess evolution acceptance. This test is a 20-item Likert-type scale instrument with six subscales of measurement (process of evolution, scientific validity, human evolution, evidence, scientific community's view, and age of the earth). This test has been considered to be internally consistent and having a high test-retest consistence [19], although certain authors have criticized it (see for example [34,35]). The original description reported only one factor, by factor analysis, which estimates evolution acceptance [28], but more recent analyses suggested that it could have two-dimensional constructs [34,36]: Facts and Credibility. Therefore, we calculated MATE scores following the original description and also using, separately, Facts and Credibility scores, following Romine et al. [34], to check whether using one or two dimensions could affect the results observed.

We chose the KEE test [41] to measure evolutionary knowledge. This is a simple test, a 10-item instrument. A few arguments support this choice. First, we needed a simple test that did not require much time during completion of the questionnaires, as we are asking teachers to use 15 minutes from a routine class. Moreover, we tried to avoid using two consecutive long tests in order to maximize student response. Furthermore, KEE and MATE had been chosen together as a practical combination in several studies [21,41,44].

## Validation and controls

We estimated the reliability of MATE and KEE data by using Cronbach's alpha statistics, as has been suggested to be done routinely for this kind of study [28,33,34]. In addition, we

consider that both our religiosity and evolution knowledge (KEE) measures needed to independently validate whether they show a high correlation with alternative constructed measurements. An 8-item questionnaire with a Likert 1–5 scale on knowledge of the theory of evolution (CTE), built and validated by Cofré et al. [45], was used for validation. For the measurement of intrinsic religiosity -IR-, the DUREL test [46,47] was used (3 items with a Likert 1–5 scale). This was accomplished in a set of 268 students from 3 degrees of the University of Granada (*n* = 264 after eliminating incomplete responses). The sample is thus composed of 220 women (83.3%) and 44 men (16.7%), which is consistent with the sex distribution in the grades. The distribution by study areas was 171 students in Social education (64.8%), 33 in Archeology (12.5%), and 60 in Criminology (22.7%). The baccalaureate (high school) curriculum was mostly that of Social Sciences (53.0%), followed by humanities (22.0%), and sciences (16.0%), while those of technology (7.2%) and arts (2.6%) were in the minority. The academic level of the participants—prior to starting the degree—was Baccalaureate (89.4%) or Bachelor/Other degree (10.2%). Regarding religiosity, 82.8% of the participants said they were not religious, while 17.2% declared themselves religious, although only 2.6% indicated they were practicing (question about religiosity associated with the MATE / KEE questionnaires). The questionnaires were administered using pencil-and-paper and later stored on Excel® sheets.

We used an indirect method to infer the relative importance of NOS understanding on MATE/KEE scores by comparing the relative value of these scores on different degrees. A high level of acceptance associated with both Chemistry and Biology would suggest a NOS (understanding) factor contributing to evolution acceptance because these two degrees share a similar science itinerary in undergraduate and degree curriculum [48], while a primacy of Biology students over the rest of students would favor an interpretation based on evolutionary knowledge because this degree is the only one to have evolutionary knowledge in the degree curriculum (S2 Table).

Additionally, we checked the level of MATE and KEE in Biology teachers from two universities (Autónoma de Madrid and Vigo). This was considered as a positive control, because we expected that the teacher level, on average, would be higher than in corresponding third-year Biology students from their own universities. A few of the professors were instructing in other degrees (Chemistry, Mathematics, etc.).

Finally, we investigated the amount of evolutionary concepts described in the Biology curricula of the same ten Universities included in the study. The evolutionary subjects relative to the number of themes per lecture and University are described in S2 Table. These allowed us to estimate the contribution of evolutionary theory to the credits degree (Evolution Credits) as the sum of the lecture credits (Credits ECTS) weighted by percentage of evolutionary concepts (Probability of Evolution content). Notice that all Spanish public degrees in Biology show 240 ECTS credits.

## Data analysis and statistics

The Google Form platform automatically generates a spreadsheet containing each participant answer in a different column from the same row. Using a homemade R script [49], we accessed each spreadsheet (one per university) and encoded the answers appropriately for downstream statistical analysis in csv files (data available at Figshare Repository; DOI:10.6084/m9.figshare.12623747). We merged all csv files and excluded questionnaires in which any student had not answered all items.

The associations among demographic variables were studied by a nonparametric chi-squared test. The probability was adjusted by the SGoF multitest procedure [50]. The main interest of the study was, however, to investigate the contribution of factors *Religiosity* (fixed),

*Degree* (fixed), and *University* (nested within the interaction) into different dependent variables (MATE; MATE Facts, MATE Credibility and KEE scores) by a three-way ANOVA. This design allowed us to focus on the factors *Religiosity* and *Degree*, being tested against their interaction pooled with the nested factor (because the *p*-value of interaction was $\geq 0.25$; [51]). This latter approach enabled us to detect if differences between levels of factors (Religiosity and Degree) persisted across the *University* variation [52]. The homoscedasticity of the data was evaluated by the Levine test. However, we used general linear models (with sum square type III), which are one of the most robust procedures to analyze data showing deviation from the parametric assumptions like normality or heteroscedasticity [53]. When the heteroscedasticity could not be corrected we checked whether the treatment with the most different mean showed the highest variance, the only dangerous heterocedasticity when interpreting ANOVA results [51].

The positive control on Biology teachers from two universities was accomplished by a two-way ANOVA on dependent variables (MATE and KEE) and factors *University* (fixed) and *Religiosity* (fixed). In this case, the correlation between *Religiosity* and MATE was accomplished within University. On the other hand, we also measured the correlation between the averaged Evolution themes within Biology curricula and the KEE score among universities. In all former cases, we used Pearson's r coefficient to estimate the degree of association, and the non-parametric Spearman r correlation to test its significance.

In addition, we used stepwise multiple linear regression to explain the evolution acceptance (MATE, MATE Facts, and MATE Credibility) by the potential exploratory variables available (*Degree*, *Religiosity*, *Age*, *Level*, and KEE scores). The variables Degree and University were recoded for this analysis following its averaged value for MATE (e.g. for *Degree*; English = 1; Chemistry = 2; History = 3; Biology = 4). We understand that such variables can confound many causes that explain their variation, but we wanted to check if they could obscure or not the other exploratory variables. The significance was checked by a regression ANOVA and the contribution of any variable entering into the model, independently of other variables, was estimated by the partial correlation coefficient [52]. All statistical analyses were done by the SPSS/PC software, version 24.0.0.0 [54], while the graphics were generated by the R project [49].

## Results

### Evolution acceptance review in the literature

We found data on MATE scores for undergraduate students in 6 countries (Table 1). The per-country average ranged from 55.5 in Pre-service education in Biology from Turkey to 84.2 in Germany.

### Validation with different tests

Cronbach's alpha gave a high reliability value for the DUREL (IR) construct (0.797), while it was lower for the CTE evolutionary knowledge one (0.635). This represents a low reliability, although at acceptable levels. Regarding the sociodemographic variables, the test of independence between pairs of variables (*Sex*, *Religiosity*, *Academic itinerary*, and *Degree*) were significant with the chi-square test only in some pairs, e.g. *Sex* and *Degree* (2 of 8 tests) and *Academic Itinerary* and *Degree* (10 of 15 tests). It should be noted that the proportion of women in the degrees in which the questionnaires were applied was very high. Regarding knowledge of the theory, the application of the CTE test yielded an average score of 26.53 (*n* = 264; Min. 18; Max. 40; SD = 3.225) which, adjusted to the range of the KEE scale for comparison, indicates a

**Table 1. Review of MATE scores (mean ± SD) for undergraduate university students from different countries.**

| Country | Academic discipline | MATE | References |
|---|---|---|---|
| Germany | Pre-service education | 84.2 | [55] Großschedl et al. 2014 |
| Spain | Third course (four degrees) | 87.2 ± 8.63 | This study |
| Greece | Pre-service education | 70.9 | [56] Athanasiou & Papadopoulou 2012 |
| Pakistan | Medicine | 58.3 | [57] Yousuf et al. 2011 |
| South Korea | Pre-service education (Biology) | ~ 73.0 | [58] Ha et al. 2012 |
| Turkey | Pre-service education (Biology) | 55.7 ± 8.46 | [59–61] Deniz et al. 2008 and 2011; Deniz & Sahin 2016 |
| USA | Various | 73.6 ± 4.74 | [21,62] Walter et al. 2013; Rissler et al. 2014 |
| | Biology | 78.4 ± 10.03 | [19,34,37,63,64] Cavallo & McCall 2008; Manwaring et al. 2015; Romine et al. 2017; Sbeglia & Nehm 2018; Barnes et al. 2019 |
| | Pre-service education (Biology and Secondary Science Education) | 70.9 | [23] Glaze et al. 2014 |

value of M = 3.16 ± 0.5. This can be considered close to that obtained in English History and Philology degrees, if we take into account the random noise in the case of KEE scores.

The data obtained when applying the DUREL indicated that just over two-thirds of the participants (69.7%) did not consider themselves religious people. The scores of the level of intrinsic religiosity through the DUREL test (with Likert scale 1–5) indicated that the degree of religiosity was low (Mean DUREL ± SD = 2.02 ± 0.935; $n$ = 264). This finding could be important to consider when comparing this test with knowledge data. Disaggregating the three items of the IR test, the religious experience turned out to be at a medium low level (DUREL = 2.34 ± 0.935) and fell still lower when respondents were asked about whether religious beliefs defined their vision of life (DUREL = 1.99 ± 1.052), and even more so when asked about trying to bring religion to other matters of life (DUREL = 1.73 ± 0.990).

We compared the DUREL IR and level of knowledge of the theory of evolution (CTE), applying a nonparametric correlation test. A moderate negative correlation coefficient (rho = -0.124; $n$ = 264; P = 0.045) was obtained.

## Analyses of questionnaires

The descriptive frequencies and scores for the different demographic variables are shown in S1 Table. Pairs of different variables (*Sex*, *Religiosity*, *Academic level*, *Itinerary*, and *Degree*) per university were usually nonsignificant by the chi-square test (15 significant out of 100 tests; S3 Table). Most significant cases affected a few pairs of variables: *Sex* by *Degree* (4 of 10 tests) and *Itinerary* by *Degree* (10 of 10 tests). The frequency of treatments for demographic variables was always heterogeneous among Universities (S4 Table). This typically occurred because just one University departed from the general trend (as in *Sex*, *Degree* and *Itinerary*), but in two variables (*Degree* and *Religiosity*) no clear general trend was found.

Our data showed a high reliability of the MATE test (Cronbach's r = 0.87; $n$ = 978; items = 20) but lower reliability for the KEE test (Cronbach's r = 0.42; $n$ = 981; items = 10), indicating a safe inference at least in the MATE analyses. Results from the multifactorial ANOVAs for different dependent variables are presented in Table 2. The MATE was heteroscedastic; however, none of the transformations used allowed for correcting this; we therefore present untransformed data analyses. Data derived on this instrument showed significant differences for *Religiosity*, *Degree*, and *University* (nested factor). Heteroscedasticity did not compromise the $p$-values for *Degree*, because the treatment that differed from the rest (Biology) showed the smallest variance. However, it could potentially compromise the $p$-value for

**Table 2. Three-way ANOVA (*Religiosity*, *Degree* and *University* nested within the interaction) for different dependent variables about evolution acceptance and knowledge.**

| Dependent | Factor | SS | DF | MS | F | Probability |
|---|---|---|---|---|---|---|
| MATE | Religion | 1451.75 | 1 | 1451.75 | 14.46 | 0.0003 |
| | Faculty | 1851.03 | 3 | 617.01 | 6.15 | 0.0009 |
| | Religion x Faculty | 351.98 | 3 | 117.33 | 1.18 | 0.3248 |
| | University (Religion x Faculty) | 6673.76 | 67 | 99.61 | 1.58 | 0.0027 |
| | Error | 56535.67 | 898 | 62.96 | | |
| MATE Facts | Religiosity | 371.54 | 1 | 371.54 | 12.41 | 0.0008 |
| | Faculty | 364.18 | 3 | 121.39 | 4.06 | 0.0102 |
| | Religiosity x Faculty | 69.43 | 3 | 23.14 | 0.77 | 0.5174 |
| | University (Religiosity x Faculty) | 2025.46 | 67 | 30.23 | 1.67 | 0.0008 |
| | Error | 16233.91 | 899 | 18.06 | | |
| MATE Credibility | Religiosity | 356.57 | 1 | 356.57 | 12.02 | 0.0009 |
| | Faculty | 610.12 | 3 | 203.37 | 6.85 | 0.0004 |
| | Religiosity x Faculty | 122.83 | 3 | 40.94 | 1.40 | 0.2495 |
| | University (Religiosity x Faculty) | 1954.51 | 67 | 29.17 | 1.51 | 0.0062 |
| | Error | 17357.64 | 900 | 19.29 | | |
| KEE | Religiosity | 4.17 | 1 | 4.17 | 1.17 | 0.2838 |
| | Faculty | 177.03 | 3 | 59.01 | 16.52 | 0.0000 |
| | Religiosity x Faculty | 2.27 | 3 | 0.76 | 0.21 | 0.8926 |
| | University (Religiosity x Faculty) | 247.79 | 67 | 3.70 | 1.27 | 0.0740 |
| | Error | 2617.02 | 901 | 2.90 | | |

SS is sum of squares, DF is degrees of freedom, MS is mean squares and F is F test

*Religiosity*, as this treatment showed a rather high variance. So our mean differences in MATE between *Religiosity* treatments could be partially caused by variance differences. Interestingly, the interaction between *Religiosity* and *Degree* was not significant, suggesting that these two factors independently contribute to acceptance of evolution. The two-dimensional scores of the MATE (MATE Facts and MATE Credibility) showed the same trend and pattern of significance as the MATE (Table 2). Actually, facts and credibility showed a rather high correlation between them ($r_{Pearson} = 0.71$; *n* = 978; $P_{Spearman} < 0.001$). The results for the MATE are graphically shown in Fig 2: a high averaged evolution acceptance > 80 was observed for all degrees (mean ± SD = 87.2 ± 8.63). The figure also shows how religious individuals tend to appear in the region of lower MATE values but with a large variance in the responses.

The data were homoscedastic for the KEE; we therefore present untransformed analyses. Only the factor *Degree* was significant for the KEE (Table 2). The differences were in the order of Biology (mean ± SD for KEE = 6.5 ± 1.71) > Chemistry (5.2 ± 1.73) > History (4.8 ± 1.81) and English Philology (4.4 ± 1.66). However, the SNK test showed that only Biology differed significantly from the other treatments. Furthermore, when we studied the KEE scores exclusively within Biology, there were significant differences among Universities (F = 2.74; $DF_1 = 9$; $DF_2 = 344$; P = 0.004). Moreover, in that case there was a significant correlation between the Evolution Credits and KEE score among the ten studied Universities (r = 0.567; *n* = 10; $P_{Spearman} = 0.011$).

A detailed analysis of evolution knowledge in Biology on the KEE items revealed three different patterns across Faculties (Fig 3). One item (K2) showed a similarly high rate of response among students of any degree, while several items (K5, K7, K8 and K9) showed similar but low scores. Finally, another group (K1, K3, K4, K6 and K10) showed the expected variations among different undergraduate students, with the highest score for those studying Biology.

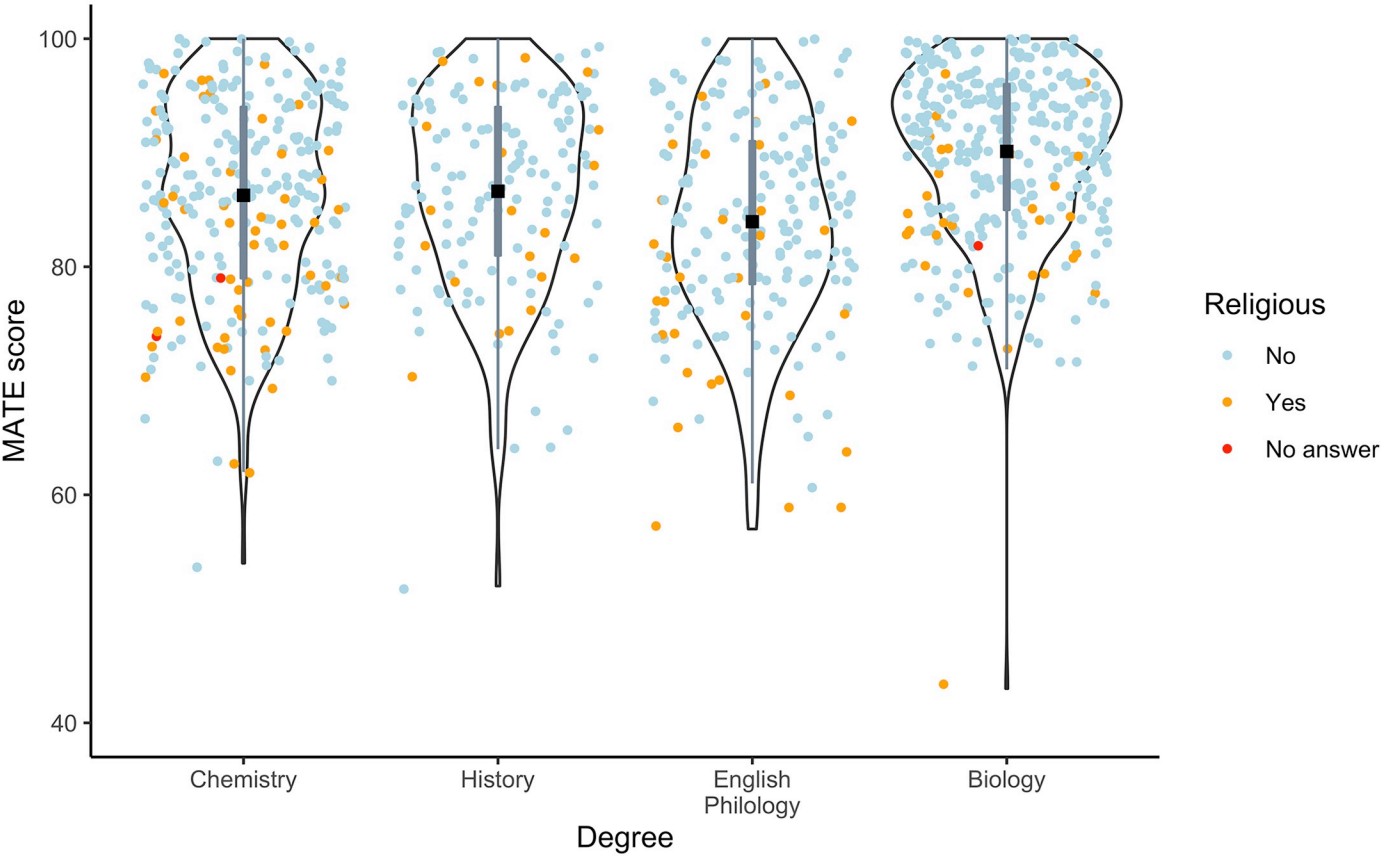

**Fig 2. Violin plots for MATE per Degree.** The individual Religiosity score of every individual is presented. MATE scores per individual are represented with dots, coloured based on their correspondent religious scores. Each violin consists of two symmetric vertical smoothed probability densities (left and right curves). Inside each violin plot there is a boxplot conformed by one narrow grey rectangle ranging from the first to the third quartile (Q1 to Q3) of the distribution and two thin lines (upper and lower whiskers) extending from Q3 to Q3 + 1.5 × (Q3—Q1) and from Q1 to Q1–1.5 × (Q3- Q1). Mean values are shown with black diamonds. Dots beyond whiskers are outliers.

A positive control of our analysis was done on the acceptance and knowledge levels of Biology teachers from two universities (Vigo and Autónoma de Madrid). In this case, the two-way ANOVA did not detect any significant factor either for the MATE or the KEE, indicating that teacher evolution acceptance and knowledge may not differ across levels of Degree and Religiosity. These dependent variables showed that teachers had a higher average than students: mean ± SD for MATE (93.6 ± 6.22; n = 50) and KEE (7.6 ± 1.86; n = 53). The correlation between Religiosity and MATE was not significant ($P_{Vigo}$ = 0.085; $P_{Madrid}$ = 0.965).

Finally, we used multiple regression analysis to explore which of the available variables (*Age*, *Religiosity*, *KEE score*, *Degree*, and *University*) could explain evolution acceptance (MATE, MATE Facts, and MATE Credibility; Table 3). The results show similar regression models under any of the acceptance scores used; in all cases, the order of variables entered into the model were KEE (evolution knowledge), followed by *Religiosity*, *University*, and *Degree*. Moreover, partial correlation coefficients were rather similar to standardized regression coefficients (not shown), which indicates that these variables do in fact contribute independently to the explanation of evolution acceptance.

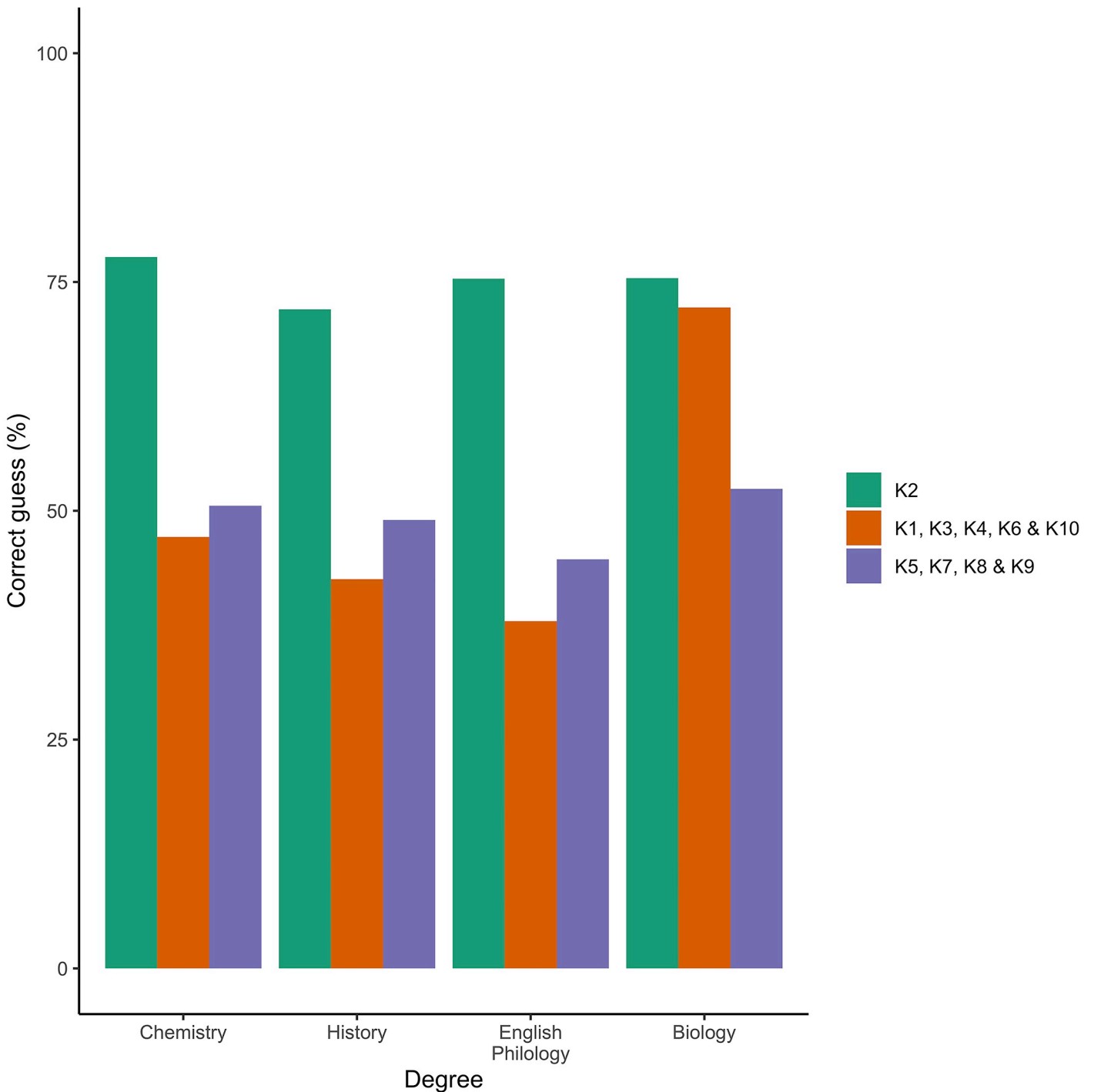

**Fig 3. Average values of correct answer per KEE item, grouped in three different classes: Most guessed (K2), most failed (K5, K7, K8 and K9) and differences between Faculties (K1, K3, K4, K6 and K10).**

**Table 3. Stepwise multiple regression of different exploratory variables (see text) to predict the different dependent variables (evolution acceptance and knowledge measures).**

| Dependent | $r^2$ | F test | Variables | Partial r |
|---|---|---|---|---|
| MATE | 0.226 | 70.8*** | KEE | 0.33*** |
| | | | Religiosity | -0.23*** |
| | | | University | 0.16*** |
| | | | Faculty | 0.11*** |
| MATE Facts | 0.182 | 54.0*** | KEE | 0.30*** |
| | | | Religiosity | -0.19*** |
| | | | University | 0.14*** |
| | | | Faculty | 0.07* |
| MATE Credibility | 0.208 | 63.6*** | KEE | 0.30*** |
| | | | Religiosity | -0.19*** |
| | | | University | 0.15*** |
| | | | Faculty | 0.14*** |

$r^2$ is squared Pearson' $r$ coefficient, $F$ test is the result of the regression ANOVA, Variables are the those included in the model in the original order (from up to down), and Partial r is the partial correlation coefficient

## Discussion

Evolutionary theory acceptance may depend on factors such as religiosity, evolution knowledge, and NOS understanding, although their relative contribution is a contentious issue [23–25]. Part of the problem is the existence of confounding factors that do not allow a focus on the actual cause (e.g. use of a range of questionnaires, often with varying structures and/or properties or the fact that different studies have been applied to different social groups). Here we have tried to focus on a well-established evolution-acceptance tool (MATE) and studied exclusively third-year students from the same four degrees at ten public Spanish Universities. This strategy may reduce the intrinsic variability of responses and allow detection of factors with relatively small contributions, compared to other studies. In relation to this, our demographic analysis suggests that only two pairs of variables were dependent: *Sex* by *Degree* and *Itinerary* by *Degree*. The first pair of variables is beyond the scope of this study, although there may exist several explanations that can influence differential academic interest in males and females (see an extensive review in Halpern [65]). The second pair of variables is an absolutely expected relationship, because pre-university itineraries are supposed to prepare the students for final degree choices. Therefore, most biology and chemistry students come from a high school itinerary in Science, while the situation is more variable in the other two degrees (although most students come from undergraduate humanities and social science itineraries).

Evolution acceptance has been considered a complex parameter to estimate. Indeed, the MATE questionnaire has been suggested by some authors [34,36] to include two constructs: Facts and Credibility. However, our separate analysis of the two construct estimators rendered similar results as were obtained with the original MATE—a result observed in other studies [34,36] as well. Therefore, irrespective of the true dimensionality of MATE, we can consider it as a valid estimate of evolution acceptance, at least in our data set.

The analysis of Spanish university students reveals that levels of evolution acceptance are relatively high (average range 81.2–92.8 per university), at the same level as most modern industrialized countries (Table 1). Actually, biology teachers from two universities obtained a higher average than their corresponding students. Interestingly, we observed differences in student's evolution acceptance for *Degree* and *Religiosity* but not for their interaction. These

observations suggest that these two factors may provide an independent contribution to the level of evolution acceptance. Actually, the same result was observed when searching an explicative model by multiple regression. We can discuss the contribution of these two factors separately.

Religiosity has been suggested to be a major contributor to evolution acceptance in several studies [18,21,35,66]. Although our religiosity estimate is somehow simpler than other published alternatives, our validation experiment confirms that the more detailed construct of the DUREL instrument rendered similar tendencies as in the general study. However, one clear limitation in our design is that we asked for religiosity irrespective of the type of religion. It could be that specific theistic views may confront evolution beliefs more directly [44]. This is something that should be corrected in future studies. It also could explain why religiosity had a relatively small influence on evolution acceptance in our data set (6.6% reduction) compared to other studies (Table 1). Moreover, in our investigation, we also detected the highest variation for MATE in the high religiosity subgroup—an association that could be explained if individuals belonging to both theist and non-theist religions could perhaps differentially contribute to it. The association of religiosity and MATE is quite variable across countries. For example, in the USA, people with higher religiosity levels can reduce its acceptance by up to 20–50% [18,41], although such reduction is typically less than 20% in biology undergraduate students (Table 1). Moreover, evolution acceptance itself may depend on education level, as in-service teachers from the USA increase levels of acceptance to 85.0 ± 4.5 [67,68]. Interestingly, the two countries with the lowest acceptance levels belong to countries with a majority of theist religious citizens, suggesting that these kinds of religions present the greatest *a priori* prejudices toward evolutionary views (Table 1). It is worth noting most controversies about these factors pertain to whether they represent a true causative factor or a correlated effect (as if high religiosity students show either low evolution knowledge or NOS levels; see Rice et al. [44]). This controversy can be partially data-dependent; thus, in certain social contexts religiosity may be a main component by itself while, in others, it could rather be indirectly mediated by other factors. Notice, however, that among Spanish university students the contribution of religiosity was relatively low but apparently independent of evolution knowledge (KEE test) or NOS level (Degree). Other authors have already shown that the factors affecting evolution acceptance differ from those contributing to the acceptance of other scientific theories [20]. In summary, a society should be warned that certain religiosities may add some prejudices to any first encounter with evolutionary theory. Perhaps our scientific community should increase efforts to show that such a negative relationship between evolution and religion is neither natural nor justified [69–71] or take an active predisposition to change it (e.g. Truong et al. [72]).

In our study, evolution acceptance also depended on the degree studied and, under the regression model, evolution knowledge. These considerations point again to the same major forces previously detected (Religiosity, Knowledge, and NOS understanding) [23–25]. Actually, the three are directly compared in the regression model; such an analysis suggests (comparing partial correlation coefficients from Table 3) that in our study the ranking of factors may be Knowledge > Religiosity > NOS. Of course, such positions may depend on the particular set of citizens studied. As we used as participants third-year university students with presumably relatively high levels of NOS, this may have influenced the results. Actually, if NOS understanding were very important, we would expect that both Biology and Chemistry students (Science itinerary) would distance themselves from History and Philology (Social Sciences and Humanities Itineraries) while, if evolution knowledge is the main driver, we would expect that Biology would succeed in isolation (as the only one with actual evolution themes). Therefore, our results suggest that, at least in our university students, evolution knowledge may be more relevant than NOS understanding.

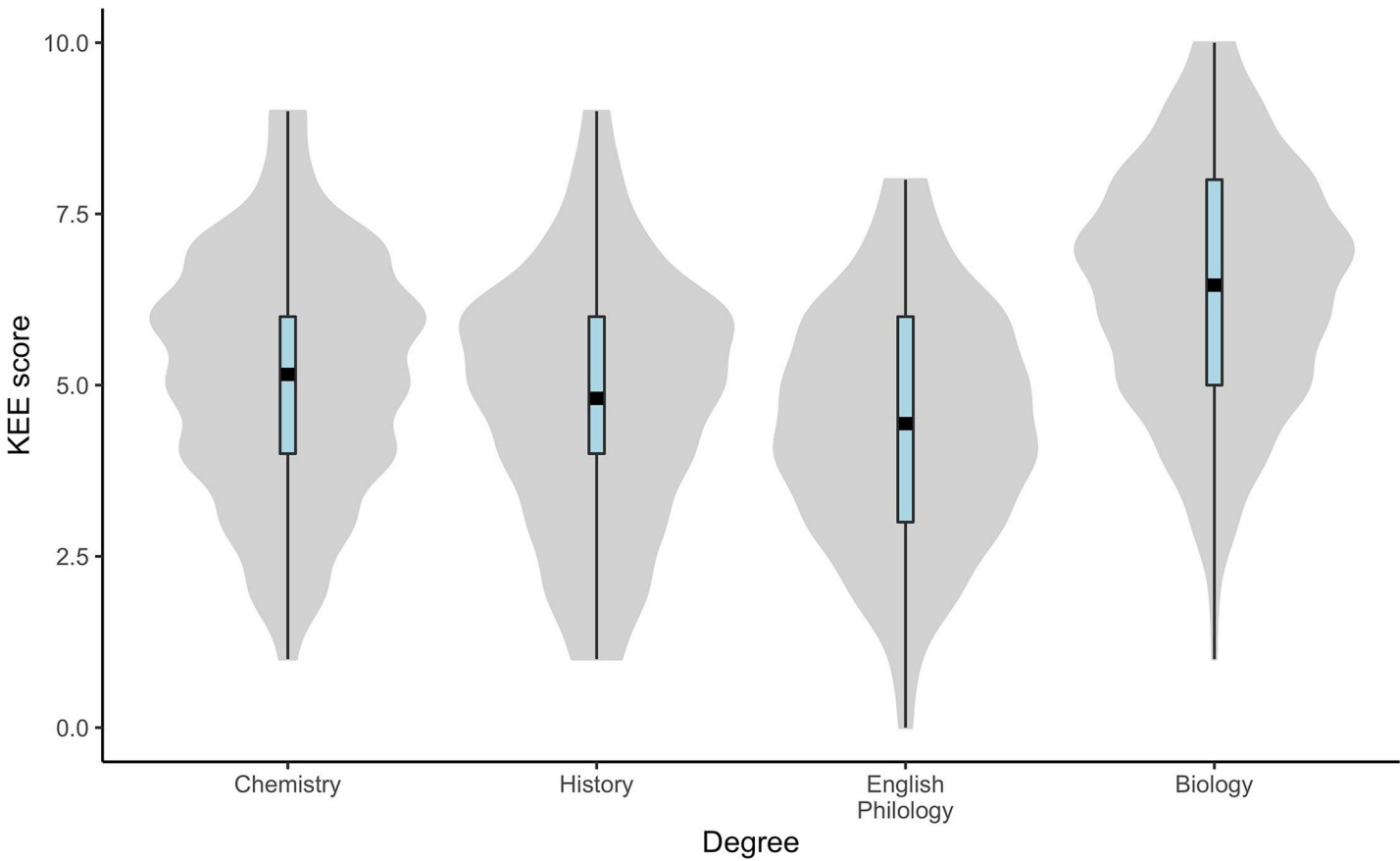

**Fig 4. Violin plots for KEE per Degree.** Violin plots and boxplots as explained in Fig 2. Here, mean values are shown with black squares and the interquartile range in blue.

A different trend was observed when evolution knowledge was evaluated by the KEE questionnaire. The analysis showed homoscedasticity for this variable across treatments; thus, all the statistical inferences were very safe. On the other hand, we found rather low levels of reliability for this test (which suggests low-quality data for KEE). This represents a limitation in our ability to interpret the evolutionary knowledge, as we may not truly understand the relationship between knowledge and acceptance given that knowledge is indeed so low. However, we believe that a relatively high proportion of individuals may answer most questions randomly (as they do not know the answer and we asked the students to answer all items from KEE). This fact may be responsible for the low reliability value observed. Indeed, this is not dependent on the item studied, as we observed the same reliability using the one-item out of the analysis strategy (not shown). For KEE, only the *Degree* factor was significant (Biology students behave better than the rest; Fig 4), suggesting that it is the amount of evolution materials in the curricula that is the most relevant factor. We have another argument that points in the same direction, the differences among universities in evolution knowledge within Biology were partially explained (up to 30%) by degree of evolutionary themes in the Biology curricula. The Evolution Knowledge observed in our students was within the range of those detected in previous studies [21,41,44]. However, in those earlier investigations, there was a relatively high variability among different social subgroups. For example, students who received the first instruction on evolution within a religious environment consistently presented a lower

evolution knowledge level (3.2) than those who had received such instruction in the high school setting (5.4; scale modified from Moore et al. [41]). On the other hand, as expected, evolution teachers from the two tested universities showed a higher knowledge level than corresponding students. In summary, in our case, all results suggested that, to improve evolution knowledge, we must introduce more evolution-related materials into the curricula. This is a well-known problem in the Spanish education system, as there is very little content on evolutionary biology in primary and secondary school curricula. Indeed, evolutionary biology contents are only available to students who choose the Science Itinerary [43]. This is something the education community should try to change in the coming years.

The study of responses for separated items for evolutionary knowledge gave different results. Item 2 was correctly answered by most students (regarding reasons for pesticide resistance; S1 Fig), while items about the level of natural selection (5 and 7), description of macro-evolution (8), and the ultimate source of variation (9) were poorly answered. The positive response to Item 2 can be interpreted to mean that most students are aware of some of the practical implications that evolutionary theory has on the living world. The items answered negatively by most students could be considered a potential guide to the kind of evolutionary concepts that are hard for the students to grasp. Other authors have focused on the differential responses item by item [40,73], although the differential tool considered may condition the result. For example, [73] Nehm and Ha [73] found that students gave more frequent correct responses under trait gain than loss, a bias that cannot be checked by the KEE.

In summary, we have observed relatively high levels of evolution theory acceptance using the MATE in third-year university students attending ten Spanish Universities. The main factors that seemed to influence levels of acceptance were *Degree* > *Religiosity* > KEE, which suggests that both religiosity and evolution knowledge are relevant and partially independent factors to be considered. A similar strategy should be accomplished in different segments of the educational course, for example, just prior to entry into the degree program or immediately after finishing it. Moreover, although the reduction in acceptance of evolution due to religiosity is relatively low, we should consider strategies to diminish the a priori prejudice against the concept, as held by highly religious individuals. Perhaps a general effort to persuade the religious part of the society that religion and evolution are not necessarily antagonistic views could be useful as well. A more serious set of limitations exists regarding evolution knowledge which, in general, has been relatively low in all grades, including biology. This suggests an obvious strategy for improving this situation, i.e. to incorporate more evolutionary theory in both pre-university levels, affecting all students, and in the university, affecting particularly the curricula in Biology and other life sciences study programs.

## Supporting information

**S1 Data. Data evolution questionaries.**
(XLSX)

**S2 Data. University questionaries teache.**
(XLSX)

**S1 Table. Descriptive parameters of the data questionnaires.** N is number of students, $N_{analysis}$ is the number used in the study (after excluding uncomplete cases). Religiosity represents the percentage of students that claim to be religious practitioner.
(DOCX)

**S2 Table. Description of the curricula of the Biology degrees per University.** Course is the year of the given lecture. Total content is the number of themes of the lecture, while subject

content is the number of those related with Evolution. Probability is the ratio between the former two, while Evolution credits is the sum of the credits x probability.
(DOCX)

**S3 Table. Pairwise chi-square tests for the different variables pairs per university.** Pink shaded for those significant (without multitest correction) and Red shaded for those significant after SGoF multitest adjustment.
(DOCX)

**S4 Table. Analysis of the homogeneity chi-square test for demographic variables within University.**
(DOCX)

**S1 Fig.**
(PDF)

## Acknowledgments

We thank all participating students as well as the Faculty Deans and teachers who helped during presentation of the online questionnaires to the students from the ten Spanish Universities. We thank the Spanish Society of Evolutionary Biology (SESBE), which allowed us to use its member list to search for participants from various Spanish regions. All authors declare they have no conflict of interest.

## Author Contributions

**Conceptualization:** Juan Gefaell, Manuel Megías.

**Data curation:** Juan Gefaell.

**Formal analysis:** Juan Gefaell, Emilio Rolán-Alvarez.

**Funding acquisition:** Emilio Rolán-Alvarez.

**Investigation:** Juan Gefaell, Mohamed Abdelaziz, Inés Álvarez, Josefa Antón, Juan Arroyo, Jose L. Bella, Anxela Bugallo, Vicente Claramonte, Emilio Lizarte, Borja Milá, Cori Ramón, Marta Vila.

**Methodology:** Juan Gefaell, Tamara Prieto.

**Project administration:** Emilio Rolán-Alvarez.

**Resources:** Tamara Prieto.

**Software:** Tamara Prieto.

**Supervision:** Emilio Rolán-Alvarez.

**Validation:** Jose L. Bella, Miguel Botella, José Gijón, Rosa M. Maroto, Manuel Megías.

**Visualization:** Emilio Rolán-Alvarez.

**Writing – original draft:** Emilio Rolán-Alvarez.

**Writing – review & editing:** Juan Gefaell, Mohamed Abdelaziz, Inés Álvarez, Josefa Antón, Juan Arroyo, Jose L. Bella, Miguel Botella, Anxela Bugallo, Vicente Claramonte, José Gijón, Emilio Lizarte, Rosa M. Maroto, Manuel Megías, Borja Milá, Cori Ramón, Marta Vila, Emilio Rolán-Alvarez.

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
