## [Decision Letter · Decision Letter 0]

28 May 2020

PONE-D-20-05773

Acceptance and knowledge of evolutionary theory among 3rd-year university students in Spain

PLOS ONE

Dear Dr. Rolan-Alvarez,

Thank you for submitting your manuscript to PLOS ONE. After careful consideration, we feel that it has merit but does not fully meet PLOS ONE’s publication criteria as it currently stands. Therefore, we invite you to submit a revised version of the manuscript that addresses the points raised during the review process.

As academic editor, I enjoyed your manuscript.  I agree that it is generally acceptable for publication.  However, I want to point out the following changes that need to be made:

I agree with Reviewer #1 that the paper is well-written in terms of structure.  However, I would recommend (as did Reviewer #2) the use of an English language editing service as there were prolific grammatical errors and awkward sentence structure throughout the manuscript that sometimes hindered the readability. The methods section needs to be clarified, specifically the sampling section. It was unclear how the bibliographic review played a role in the data collected.  I am assuming that all of the results reported were from the 1050 third course students that were actually sampled.  It was just a bit unclear.I agree with Reviewer #2 that the low reliability of the KEE is a concern.  It likely means that your respondents knew relatively little about evolution, which is a point I do believe you are trying to make.  I would just recommend that you make this point more clear and that you also include it as a limitation in your discussion, especially given that we may not truly understand the relationship between knowledge and acceptance given that knowledge is indeed so low. In addition, I did find it a major limitation that religiosity did not take into account whether they were theistic or non-theistic.  It was mentioned in the article, and I appreciate that.  But, I would also like to see that clearly enunciated in a limitations section in the discussion, as well. I do not agree with Reviewer #2 that the manuscript does not make a contribution. I believe it does make a contribution to the literature and so recommend its publication pending the addressing of these few concerns. 

We would appreciate receiving your revised manuscript by Jul 12 2020 11:59PM. To enhance the reproducibility of your results, we recommend that if applicable you deposit your laboratory protocols in protocols.io, where a protocol can be assigned its own identifier (DOI) such that it can be cited independently in the future. For instructions see: http://journals.plos.org/plosone/s/submission-guidelines#loc-laboratory-protocols

We look forward to receiving your revised manuscript.

Kind regards,

Jamie L. Jensen, Ph.D.

Academic Editor

PLOS ONE

Journal Requirements:

2. PLOS ONE does not copy edit accepted manuscripts. Please proofread for typos and grammar.

3. For this observational study, please avoid causal-sounding language (such as 'impact' or 'effect') when reporting associations.

4. Please clarify how you assessed NOS understanding, with what instrument and whether it was translated (if you developed a questionnaire as part of this study and it is not under a copyright more restrictive than CC-BY, please include a copy, in both the original language and English, as Supporting Information). Please also clarify why social science and humanities majors should be expected to have a lower NOS understanding than science majors (lines 458-461).

Please include additional information regarding the survey or questionnaire used in the study and ensure that you have provided sufficient details that others could replicate the analyses. For instance, if you developed a questionnaire as part of this study and it is not under a copyright more restrictive than CC-BY, please include a copy, in both the original language and English, as Supporting Information.

Additional Editor Comments (if provided):

Reviewers' comments:

Reviewer's Responses to Questions

**Comments to the Author**

1. Is the manuscript technically sound, and do the data support the conclusions?

Reviewer #1: Yes

Reviewer #2: Yes

2. Has the statistical analysis been performed appropriately and rigorously? 

Reviewer #1: Yes

Reviewer #2: Yes

3. Have the authors made all data underlying the findings in their manuscript fully available?

Reviewer #1: Yes

Reviewer #2: Yes

4. Is the manuscript presented in an intelligible fashion and written in standard English?

Reviewer #1: Yes

Reviewer #2: Yes

5. Review Comments to the Author

Reviewer #1: Good day and thank you for the opportunity to review your work. I found the paper to be very thoughtful and well written, making strong arguments for the need for understanding as well as the choices of instrumentation to apply and the methods undertaken. I do not have specific line item comments because I felt the paper was very well written, however, I do think it could benefit from some structural breaking into paragraphs to enhance readability and flow from concept to concept. Two things that are generally present that I did not see are implications of the study and recommendations for filling gaps in the literature that are still present. While this is elementary in nature, these are key connections that should be drawn to situate the paper well within the literature that has been generated in evolution education, understanding, and knowledge in the last two decades.

Reviewer #2: I enjoyed reading this. The sample size is good and the literature review and statistical analysis are thorough. I am afraid though that I do not feel the submission makes enough of a contribution to the international literature to warrant publication in PLOS ONE and I cannot see how this can be remedied. I suggest submission is made to a science education journal (not JRST or Science Education - IJSE might be a possibility).

The manuscript is written in good English though it would benefit from copyediting by someone whose first language is English. For example, in line 87 "orthodox religious views" might confuse some reader as it might be thought that Orthodox Christianity is being referred to.

The low reliability for the KEE test (Cronbach’s r=0.42) is a bit of a concern (and a slight surprise) but nothing can be done about this.

6. PLOS authors have the option to publish the peer review history of their article (what does this mean?). If published, this will include your full peer review and any attached files.

Reviewer #1: No

Reviewer #2: No

---

## [Author Response · Author response to Decision Letter 0]

8 Jul 2020

PONE-D-20-05773

Acceptance and knowledge of evolutionary theory among 3rd-year university students in Spain

PLOS ONE

Dear Dr. Jensen,

Thank you very much for your helpful and constructive comments on our MS. We have detailed our reply below to both your and referee’s comments in blue ink. We have provided a new complete version both highlighted and unmarked. We hope that the present version will be of enough quality for being publishable in PLOS ONE.

On behalf of the rest of coauthors and myself, Emilio Rolán-Alvarez

RESPONSE TO EDITOR AND REFEREES

EDITOR:

As academic editor, I enjoyed your manuscript. I agree that it is generally acceptable for publication. However, I want to point out the following changes that need to be made:

• I agree with Reviewer #1 that the paper is well-written in terms of structure. However, I would recommend (as did Reviewer #2) the use of an English language editing service as there were prolific grammatical errors and awkward sentence structure throughout the manuscript that sometimes hindered the readability. 

We have submitted the final version to the Proof-reading-service.com company, which has corrected the English style and grammar, according to the PLOS ONE style.

• The methods section needs to be clarified, specifically the sampling section. It was unclear how the bibliographic review played a role in the data collected. I am assuming that all of the results reported were from the 1050 third course students that were actually sampled. It was just a bit unclear.

We think that you are asking whether the evolutionary concepts described in the biology curricula of the ten universities were done on exactly the same 10 universities used to obtain the questionnaires. The answer is yes, you can check the detailed data in Supplementary Table S2. We have lightly changed the sentence where this methodology was described. We have changed the sentence to make this clearer: “Finally, we investigated the amount of evolutionary concepts described in the Biology curricula of the same ten Universities used for the questionnaires.” 

Results are indeed based on the 1050 third-year students answers. We have made this clearer by modifying the first paragraph within the Data analysis and statistics section and adding a supplementary HTML containing the code used for obtaining the data. 

• I agree with Reviewer #2 that the low reliability of the KEE is a concern. It likely means that your respondents knew relatively little about evolution, which is a point I do believe you are trying to make. I would just recommend that you make this point more clear and that you also include it as a limitation in your discussion, especially given that we may not truly understand the relationship between knowledge and acceptance given that knowledge is indeed so low. 

We have changed the sentence to do this point clear enough. “This represents a limitation in our ability to interpret the evolutionary knowledge, as we may not truly understand the relationship between knowledge and acceptance given that knowledge is indeed so low. However, we believe that a relatively high proportion of individuals may answer most questions randomly (as they do not know the answer), as we have forced to the students to answer all items from KEE, and this effect may determine the low reliability value observed.”

• In addition, I did find it a major limitation that religiosity did not take into account whether they were theistic or non-theistic. It was mentioned in the article, and I appreciate that. But, I would also like to see that clearly enunciated in a limitations section in the discussion, as well. 

We have modified a sentence in the discussion to do this point clear enough. “However, one clear limitation in our design is that we asked for religiosity irrespective of the type of religion, while it could be that specific theistic views may confront evolution views more directly (Rice et al. 2015). This is something that should be corrected in future studies.”

• I do not agree with Reviewer #2 that the manuscript does not make a contribution. I believe it does make a contribution to the literature and so recommend its publication pending the addressing of these few concerns. 

Thank you for positive feed-back.

  

RESPONSE TO REVIEWERS

Reviewer #1: Good day and thank you for the opportunity to review your work. I found the paper to be very thoughtful and well written, making strong arguments for the need for understanding as well as the choices of instrumentation to apply and the methods undertaken. I do not have specific line item comments because I felt the paper was very well written, however, I do think it could benefit from some structural breaking into paragraphs to enhance readability and flow from concept to concept. Two things that are generally present that I did not see are implications of the study and recommendations for filling gaps in the literature that are still present. While this is elementary in nature, these are key connections that should be drawn to situate the paper well within the literature that has been generated in evolution education, understanding, and knowledge in the last two decades.

Thank you for the positive and constructive suggestion. We added a new sentence in the final paragraph of discussion: “A similar strategy should be accomplished in different periods of the educational system, for example just previous to entry into the degree or immediately after finishing it. Moreover, although the reduction in evolution acceptance due to religiosity is relatively low, we should consider strategies to diminish the a priori prejudice about evolution found among religious practitioners.”

Reviewer #2: I enjoyed reading this. The sample size is good and the literature review and statistical analysis are thorough. I am afraid though that I do not feel the submission makes enough of a contribution to the international literature to warrant publication in PLOS ONE and I cannot see how this can be remedied. I suggest submission is made to a science education journal (not JRST or Science Education - IJSE might be a possibility).

The manuscript is written in good English though it would benefit from copyediting by someone whose first language is English. For example, in line 87 "orthodox religious views" might confuse some reader as it might be thought that Orthodox Christianity is being referred to.

We changed this sentence to: “Such particularities have been attributed to the fact that certain rigid religious views clash with evolutionary views, together with the variation in religiosity observed among countries or even social groups within country (Coyne 2009; Rissler et al. 2014; Gallup 2017).”

The low reliability for the KEE test (Cronbach’s r=0.42) is a bit of a concern (and a slight surprise) but nothing can be done about this.

We have added one sentence explaining that this is a limitation of our study.

 RESPONSE TO JOURNAL REQUIREMENTS

We have followed all Journal requirements during submission.

2. PLOS ONE does not copy edit accepted manuscripts. Please proofread for typos and grammar.

We have send the MS to a company for proofreading.

3. For this observational study, please avoid causal-sounding language (such as 'impact' or 'effect') when reporting associations.

We have used influence, noise, or other synonymous instead of impact or effect when convenient.

4. Please clarify how you assessed NOS understanding, with what instrument and whether it was translated (if you developed a questionnaire as part of this study and it is not under a copyright more restrictive than CC-BY, please include a copy, in both the original language and English, as Supporting Information). Please also clarify why social science and humanities majors should be expected to have a lower NOS understanding than science majors (lines 458-461).

Regarding NOS understanding we explain in the text how we infer its relevance (lines 240-247).

Please include additional information regarding the survey or questionnaire used in the study and ensure that you have provided sufficient details that others could replicate the analyses. For instance, if you developed a questionnaire as part of this study and it is not under a copyright more restrictive than CC-BY, please include a copy, in both the original language and English, as Supporting Information.

All our questionnaires have been already published and so it has no sense to provide any copy of them. We provide enough information in the MS to replicate the study.

The data is available at Figshare Repository; DOI:10.6084/m9.figshare.12623747.

---

## [Editor Report · Decision Letter 1]

17 Aug 2020

Acceptance and knowledge of evolutionary theory among third-year university students in Spain

PONE-D-20-05773R1

Dear Dr. Rolan-Alvarez,

We’re pleased to inform you that your manuscript has been judged scientifically suitable for publication and will be formally accepted for publication once it meets all outstanding technical requirements.

Kind regards,

Jamie L. Jensen, Ph.D.

Academic Editor

PLOS ONE

Additional Editor Comments (optional):

I commend the authors for their efforts in improving the manuscript.  The English has been greatly improved.  I did find just a few areas that need attention (Regarding general grammar and understanding):

Line 127:  ‘evolutionary’ should  be ‘evolution’Line 230-231: a bit confusing.  I’m not sure what you mean by it.Line 346: consider: “…suggesting that these two factors independently contribute to acceptance of evolution.”Line 375: consider replacing “pointing” with “indicating”Line 376: “These dependent variables showed that teachers had a higher average than students.” Is that what you are trying to say? It is a bit confusing.Throughout the manuscript: when referring to NOS, what you are really referring to is an understanding of NOS. For example, in the first sentence of the discussion, you indicate that acceptance is likely influenced by knowledge of evolution, religiosity, and NOS. But, it shouldn't be influenced by the nature of science but rather by a participant’s understanding of the nature of science.  Change throughout.Line 439:  consider revising to “…people with higher religiosity levels have a decreased acceptance by up to 20-50%...”Line 442:  similar issue.  Revise to “…graduated (replace with in-service?) teachers from the USA have higher acceptance of 85.0…”Line 525:  “Zealots” has a very negative connotation and therefore makes your statement harsher than I think you intend it.  Perhaps consider replacing with “highly religious individuals”
---

## [Editor Report · Acceptance letter]

20 Aug 2020

PONE-D-20-05773R1 

Acceptance and knowledge of evolutionary theory among
third-year university students in Spain 

Dear Dr. Rolan-Alvarez:

I'm pleased to inform you that your manuscript has been deemed suitable for publication in PLOS ONE. Congratulations! Your manuscript is now with our production department. 

Kind regards, 

on behalf of

Dr. Jamie L. Jensen 

Academic Editor

PLOS ONE